# Implications of Heterogeneity of Epithelial-Mesenchymal States in Acromegaly Therapeutic Pharmacologic Response

**DOI:** 10.3390/biomedicines10020460

**Published:** 2022-02-16

**Authors:** Joan Gil, Montserrat Marques-Pamies, Elena Valassi, Araceli García-Martínez, Guillermo Serra, Cristina Hostalot, Carmen Fajardo-Montañana, Cristina Carrato, Ignacio Bernabeu, Mónica Marazuela, Helena Rodríguez-Lloveras, Rosa Cámara, Isabel Salinas, Cristina Lamas, Betina Biagetti, Andreu Simó-Servat, Susan M. Webb, Antonio Picó, Mireia Jordà, Manel Puig-Domingo

**Affiliations:** 1Endocrine Research Unit, Germans Trias i Pujol Research Institute (IGTP), 08916 Barcelona, Spain; jgil@igtp.cat (J.G.); hrodriguez@igtp.cat (H.R.-L.); 2Research Center for Pituitary Diseases, Department of Endocrinology/Medicine, Hospital Sant Pau, Universitat Autònoma de Barcelona, 08041 Barcelona, Spain; evalassi@santpau.cat (E.V.); swebb@santpau.cat (S.M.W.); 3Department of Endocrinology and Nutrition, Germans Trias i Pujol University Hospital, 08916 Barcelona, Spain; mmarques.germanstrias@gencat.cat (M.M.-P.); isalinas.germanstrias@gencat.cat (I.S.); 4Department of Endocrinology & Nutrition, Institute for Health and Biomedical Research (ISABIAL), Hospital General Universitario de Alicante, 03010 Alicante, Spain; bc2gamma@uco.es (A.G.-M.); antonio.pico@umh.es (A.P.); 5Biomedical Research Networking Center in Rare Diseases (CIBERER), Institute of Health Carlos III (ISCIII), 28029 Madrid, Spain; 6Department of Endocrinology, Son Espases University Hospital, 07120 Palma de Mallorca, Spain; guillermo.serra@ssib.es; 7Department of Neurosurgery, Germans Trias i Pujol University Hospital, 08916 Barcelona, Spain; chostalot.germanstrias@gencat.cat; 8Endocrinology Department, Hospital Universitario de La Ribera, 46600 Valencia, Spain; fajardo_carmon@gva.es; 9Department of Pathology, Germans Trias i Pujol University Hospital, 08916 Barcelona, Spain; ccarrato.germanstrias@gencat.cat; 10Endocrinology Division, Complejo Hospitalario Universitario de Santiago de Compostela (CHUS)-SERGAS, 15706 Santiago de Compostela, Spain; ignacio.bernabeu.moron@sergas.es; 11Department of Endocrinology, Hospital de la Princesa, Instituto Princesa, Universidad Autónoma de Madrid, 28006 Madrid, Spain; monica.marazuela@salud.madrid.org; 12Endocrinology Department, Hospital Universitario y Politécnico La Fe, 46026 Valencia, Spain; camara_ros@gva.es; 13Department of Endocrinology and Nutrition, Hospital General Universitario de Albacete, 02006 Albacete, Spain; clamaso@sescam.jccm.es; 14Department of Endocrinology, University Hospital Vall d’Hebron, 08035 Barcelona, Spain; bbiagetti@vhebron.net; 15Department of Endocrinology, Hospital Universitari Mutua Terrassa, 08221 Terrassa, Spain; asimo@mutuaterrassa.cat; 16Department of Clinical Medicine, Miguel Hernandez University, 03202 Elche, Spain; 17Department of Medicine, Autonomous University of Barcelona, 08913 Barcelona, Spain

**Keywords:** epithelial-mesenchymal transition, acromegaly, somatostatin receptor ligands (SRLs), somatostatin analogs (SSAs), RORC, SNAI1, presurgical SRLs treatment, pituitary tumors

## Abstract

Acromegaly is caused by excess growth hormone (GH) produced by a pituitary tumor. First-generation somatostatin receptor ligands (SRLs) are the first-line treatment. Several studies have linked E-cadherin loss and epithelial-mesenchymal transition (EMT) with resistance to SRLs. Our aim was to study EMT and its relationship with SRLs resistance in GH-producing tumors. We analyzed the expression of EMT-related genes by RT-qPCR in 57 tumors. The postsurgical response to SRLs was categorized as complete response, partial response, or nonresponse if IGF-1 was normal, had decreased more than 30% without normalization, or neither of those, respectively. Most tumors showed a hybrid and variable EMT expression profile not specifically associated with SRL response instead of a defined epithelial or mesenchymal phenotype. However, high *SNAI1* expression was related to invasive and SRL-nonresponsive tumors. *RORC* was overexpressed in tumors treated with SRLs before surgery, and this increased expression was more prominent in those cases that normalized postsurgical IGF-1 levels under SRL treatment. In conclusion, GH-producing tumors showed a heterogeneous expression pattern of EMT-related genes that would partly explain the heterogeneous response to SRLs. *SNAI1* and *RORC* may be useful to predict response to SRLs and help medical treatment decision making.

## 1. Introduction

Acromegaly is a rare disorder mainly caused by a growth hormone (GH)-producing pituitary tumor. The main symptoms and severe comorbidities are produced by persistent GH excess and consequently high levels of IGF-1. Other symptoms are derived from compression of the pituitary or surrounding structures [1]. First-line treatment includes endoscopic surgery [2], but the surgical cure rate is around 50% in patients presenting extrasellar growth [3]. In those cases, patients receive adjuvant therapy with first-generation somatostatin receptor ligands (SRLs) [4,5]. However, the response to SRLs varies widely among the different studies (success rates of 20–40%) [5,6,7], and the resulting comorbidities of an uncontrolled disease severely impacts patients’ lives [1,8,9]. Several markers have been identified to predict SRL resistance to avoid months of unsuccessful treatment in unresponsive patients, but most of them have been validated individually, and their predictive power varies across studies [7,10,11]. Recently, we analyzed almost all previously reported molecular markers in a large series of GH-producing pituitary tumors, and confirmed that the progressive loss of response to SRLs was associated with the loss of E-cadherin [12].

Epithelial cadherin, or E-cadherin, is a calcium-dependent cell–cell adhesion protein [13] whose loss is associated with advanced-stage cancer and poor prognosis in solid tumors derived from epitheliums [14]. Moreover, the loss of E-cadherin is a pivotal event of epithelial-mesenchymal transition (EMT), a process that consists of the transdifferentiation of epithelial cells into mesenchymal cells [15]. This mechanism allows epithelial cells to acquire invasive properties [16]. Several works have demonstrated the association of EMT [17,18], and specifically the loss of E-cadherin [19,20,21], with larger tumors and invasion, as well as lack of response to SRLs in GH-producing tumors [22]. Epithelial Splicing Regulator 1 (*ESRP1*) has been proposed to be a master regulator of EMT in somatotropinomas by modifying splicing programs [17,18]. SRLs’ resistance and EMT have been associated with AIP-mutated tumors [23,24].

In this work, we aimed to study EMT in GH-producing tumors and its relation with SRL treatment by evaluating the expression of EMT-related genes to provide SRL response predictors that allow a more personalized approach for acromegaly patients [11].

## 2. Materials and Methods

### 2.1. Patients

A total of 57 acromegaly patients from the REMAH cohort [25] recruited from 15 Spanish tertiary centers who underwent pituitary surgery and were not cured were included in the study. The phenotypic characteristics of the cohort are presented in Table 1. In the present study, all patient with a sufficient residual sample were reevaluated for EMT characterization [12].

The cohort of patients was intended to reflect the daily practice of acromegaly management. Of the 57 patients, 46 (80.7%) received first-generation SRL treatment (octreotide or lanreotide) before surgery, while 11 (19.3%) did not (Figure 1). SRL pretreatment was given for a median of seven months before surgery, aiming to improve the outcome of the surgery. All patients categorized for SRL postsurgical response were treated for at least 6 months after surgery under maximal effective therapeutic (octreotide or lanreotide) doses according to IGF-1 decrease. Follow-up data after surgery in relation to SRLs response were available for 43 patients, 38 of which had received SRL treatment presurgically. Regarding SRL response, patients were categorized as complete responders (CR) if IGF-1 was normalized, partial responders (PR) if IGF-1 decreased >30% from basal status without normalization, or nonresponders (NR) if IGF-1 decrease was <30% without normalization [26]. To better characterize biologically these patients, *GNAS* mutation was assessed as described in Puig-Domingo et al. [12]. A total of 39 tumors (68%) presented extrasellar extension of the tumor: 3 patients (5%) presented only infrasellar extension, 20 (35%) presented only suprasellar extension, and 16 (28%) presented both. Suprasellar invasion was defined as a clear tumor growth through the diaphragma sella or above the plane of the inferior optic chiasm. Infrasellar invasion was defined as a clear tumor growth through the sellar floor and into the sphenoid sinus or clivus [27].

### 2.2. DNA and RNA Isolation

Representative fragments of the fresh tumor were selected by a pathologist and embedded in RNAlater (Invitrogen, Carlsbad, CA, USA) for 24 h, after which the samples were frozen at −80 °C. Histologic examination of the frozen samples performed by a pathologist confirmed that they were part of the adenoma. DNA and RNA extraction were performed using an AllPrep DNA/RNA/miRNA Universal Kit (Qiagen, Hilden, Germany). We removed contaminating genomic DNA from RNA. Samples were treated with RNAse-free DNAse twice; first, during the extraction of RNA following the manufacturer’s protocol; and second, before the retrotranscription, for which ezDNase Enzyme (Invitrogen, Carlsbad, CA, USA) was used. The quantity and purity of DNA and RNA were measured using a NanoDrop™ 1000 spectrophotometer (RRID:SCR_016517, Thermo Fisher Scientific, Waltham, MA, USA).

### 2.3. E-Cadherin Promoter Methylation Assessment

First, 300 ng of DNA was bisulfite-converted using the EZ DNA Methylation-Gold Kit (Zymo Research, Irvine, CA, USA) according to the manufacturer’s instructions. We used bisulfite-converted DNA as a template for a nested PCR. The sequence of external primers was: 5′-GATTTTAGGTTTTAGTGAGTT-3′ (sense) and 5′-CCTACAACAACAACAACA-3′ (antisense), with an annealing temperature of 50 °C and amplified a 447 bp product. The sequence of internal primers was: 5′- GTAATTTTAGGTTAGAGGG-3′ (sense) and 5′-CTCCAAAAACCCATAACT-3′ (antisense), with an annealing temperature of 50 °C and amplified with a 321 bp product. We used the IMMOLASE DNA Polymerase (Bioline USA Inc., Memphis, TN, USA) following the manufacturer’s protocol. We processed the samples in duplicate to ensure a more representative methylation profile. The PCR program was: 10 min at 95 °C, initial polymerase activation; 30 s at 94 °C, 30 s at 50 °C, and 30 s at 72 °C for 25 cycles for the external PCR and for 35 cycles for the internal PCR; and 8 min at 72 °C for the final elongation. The duplicates were pooled, purified using the Exonuclease I and FastAP Thermosensitive Alkaline Phosphatase system (Thermo Fisher Scientific, Waltham, MA, USA), and analyzed by Sanger sequencing (Eurofins Genomics, Ebersberg, Germany).

We calculated the percentage of methylation by comparing the peak height of the cytosine residues with the peak of the thymine residues (C/ (C + T) × 100). Ranges of DNA methylation, specifically 0–10%, 11–25%, 27–50%, 51–75%, and 76–100%, for each CpG were considered. Representation of the methylation values was done using the Methylation Plotter (available at: http://maplab.cat/methylation_plotter, accessed on 15 November 2021) [28].

### 2.4. Reverse Transcription and Quantitative Polymerase Chain Reaction (RT-qPCR)

Reverse transcription was performed on 500 ng of total RNA using SuperScript IV reverse transcriptase (Invitrogen, Carlsbad, CA, USA) and random hexamers in a final volume of 20 uL according to the manufacturer’s protocol. We used Taqman assays (Applied Biosystems, Foster City, CA, USA) for quantifying gene expression, and quantitative polymerase chain reactions (qPCR) were carried out in a 7900HT Fast Real-Time PCR System (Applied Biosystems, Fosters City, CA, USA).

The genes analyzed were: Snail Family Transcriptional Repressor 1 (*SNAI1*, Hs00195591_m1), Snail Family Transcriptional Repressor 2 (*SNAI2*, Hs00950344_m1), Epithelial Splicing Regulatory Protein 1 (*ESRP1*, Hs00214472_m1), RAR Related Orphan Receptor C (*RORC*, Hs01076112_m1), N-cadherin (*CDH2*, Hs00983056_m1), Twist Family bHLH Transcription Factor 1 (*TWIST1*, Hs00361186_m1), and Vimentin (*VIM*, Hs00958111_m1). We used the previously obtained expression data for E-cadherin (*CDH1*) in the analysis of correlations and the unsupervised hierarchical clustering [12].

We used TaqMan Gene Expression Master Mix (Applied Biosystems, Foster City, CA, USA), and we performed triplicate reactions for each sample in a final volume of 10 uL in 384-well plates. All genes for each sample were analyzed in the same plate in order to minimize the interassay variation and quantify relative gene expression. We calculated a normalization factor for each sample based on the geometric mean of three reference genes (TATA-Box Binding Protein, *TBP*, Hs00427621_m1; Mitochondrial Ribosomal Protein L19, *MRPL19*, Hs01040217_m1; and Phosphoglycerate Kinase 1, *PGK1*, Hs00943178_g1) in order to quantify relative gene expression following geNorm guidance algorithms (RRID:SCR_006763; https://genorm.cmgg.be, accessed on 1 December 2020) [29].

### 2.5. RORC Immunohistochemistry

We cut formalin-fixed, paraffin-embedded tumor samples into sequential 4 µm thick sections. Subsequently, we stained the tissue sections using a fully automated Ventana BenchMark ULTRA stainer (Ventana, Tucson, AZ, USA) according to the manufacturer’s instructions. Diaminobenzidine was used as a substrate for binding of peroxidase-coupled antibodies. Moreover, the sections were counterstained with hematoxylin. The mouse monoclonal anti-ROR gamma antibody clone 6F3.1 (MABF81, Merck Millipore, Burlington, MA, USA) was used at a dilution of 1:50. Normal hepatic tissue was used as a positive control for staining. Immunostaining of *RORC* was classified according to the immunopositivity of the tumor nuclei.

### 2.6. Biochemical and Hormonal Assays

Blood samples were collected after an overnight fasting at baseline and at different follow-up time points. IGF-1 was quantified through 2 different methodologies. Method 1 used an Immunotech IGF1 kit (Immunotech-Beckman, Marseille, France). Expected values depending on age were: 20–30 y, 220–550 ng/mL; 30–40 y, 140–380 ng/mL; 40–50 y, 54–330 ng/mL; and 50–60 y, 94–285 ng/mL. The intra-assay CV was less than 6.3%; inter-assay CV, 6.8%; and sensitivity, 30 ng/mL. Method 2 used a Diagnostic Systems Laboratories nonextraction immunoradiometric assay (Diagnostic Systems Laboratories Inc, Webster, TX, USA). We calculated the theoretical sensitivity, or minimum detection limit, by interpolation of the mean plus two SD values of 20 replicates of the 0 ng/mL. IGF-1 standard was 2 ng/mL. The interassay CVs were 7.4 and 4.2 for the concentrations of 32.5 and 383.8 ng/mL, respectively. The interassay CVs were 7 and 3.9 for the mean concentration values of 34.03 and 373.86 ng/mL, respectively.

### 2.7. Statistical Analysis

Numerical variables were expressed as mean ± standard error or standard deviation, as appropriate. We investigated the potential identification of patient response subgroups using an unsupervised hierarchical clustering based on their molecular expression profile. The cluster criterion was determined with Ward’s method, and the distances were calculated with the Manhattan method. We performed Pearson’s correlations for all quantitative variables. The differences between groups were tested using analysis of variance (Student’s *t*-test, Wilcoxon signed-rank test, or Kruskal–Wallis analysis of variance, as appropriate). We constructed a binomial logistic regression model between different SRL response categories to determine the differences in each normalized gene expression. We adjusted the model by age and gender. We assessed the classification power of the model through a receiver operating characteristic (ROC) curve analysis. All *p*-values were two-sided, and statistical significance was considered when *p* < 0.05. All statistical analyses were performed using R version 3.3.3 (R Project for Statistical Computing, RRID:SCR_001905). Unsupervised hierarchical clustering was performed using the R package pheatmap (Pretty Heatmaps, https://CRAN.R-project.org/package=pheatmap, accessed on 22 April 2021). The graphical representation was done using package ggplot 2 (RRID:SCR_014601, Whickham, https://CRAN.R-project.org/package=ggplot2, accessed on 30 December 2021), and the *p*-values were added using ggpubr package (‘ggplot2’ Based Publication Ready Plots, https://CRAN.R-project.org/package=ggpubr, accessed on 30 December 2021). Correlation plots were done using corrplot package (http://cran.r-project.org/package=corrplot, accessed on 30 December 2021), and ROC curves were plotted using pROC package (Display and Analyze ROC Curves, https://CRAN.R-project.org/package=pROC, accessed on 30 December 2021).

## 3. Results

### 3.1. E-Cadherin Expression Did Not Correlate with Promoter Methylation

The epigenetic silencing of E-cadherin by the hypermethylation of its promoter occurs in a wide variety of tumor types [30]. To investigate the loss of E-cadherin expression in the SRL nonresponder GH-producing tumors [12], we analyzed the DNA methylation of E-cadherin promoter in 10 tumors with extreme levels of E-cadherin expression, 5 with low expression, and 5 with high expression. Results showed that the promoter was unmethylated in all samples, whether the expression of E-cadherin was high or low (Figure 2).

### 3.2. SRLs Treatment before Surgery Affected the Expression of Some EMT-Related Genes

The loss of response to SRLs associated with the loss of E-cadherin in GH-producing tumors suggested a relationship between EMT and the response to SRLs. Thus, we analyzed the expression of a subset of EMT-related genes, including the epithelial marker *ESRP1*; the mesenchymal markers vimentin, N-cadherin, *SNAI1*, *SNAI2*, and *TWIST1* (the last three being transcription factors); and *RORC*, which previously has been related to E-cadherin and EMT in acromegaly [31]. Gene expression levels were evaluated in patients presurgically treated with SRLs and compared with those not receiving this treatment (*n* = 46 and *n* = 11, respectively). *RORC* and N-cadherin showed higher expression levels in tumors presurgically treated (*p* = 0.004 and *p* = 0.017, respectively), with RORC showing an increase of higher magnitude than the one observed for N-cadherin (Figure 3). Expression of Vimentin, *SNAI1*, *SNAI2*, *TWIST1*, and *ESRP1* did not show significant differences between tumors presurgically treated with SRLs or naïve tumors (data not shown).

The other genes did not show any change by presurgical SRLs treatment. However, the analysis of gene expression correlations according to whether patients were treated or not with SRLs before surgery showed different behaviors. In patients not treated presurgically, we found strong positive correlations between epithelial genes and between mesenchymal genes, as well as strong negative correlations between epithelial and mesenchymal genes; while in pretreated patients, these correlations were reduced or even lost (Figure 4), implicating EMT in the GH-producing tumor phenotype. Interestingly, *RORC* correlated positively with epithelial genes and negatively with mesenchymal genes in non-pretreated patients, suggesting an association of *RORC* with EMT.

### 3.3. Association of EMT Markers with Clinical Variables

We analyzed the correlation of each molecular marker with the different tumor characteristics (Appendix A). In the case of N-cadherin and *RORC*, whose expression is affected by presurgical SRL treatment, the analysis was performed by segregating pretreated and non-pretreated patients. Tumor size was related to N-cadherin both in pretreated and non-pretreated patients (*p* = 0.034 and 0.047, respectively). Interestingly, we found higher levels of *SNAI1* in tumors with extrasellar extension (*p* = 0.049) (Figure 5). We also found an association between *GNAS* mutational status and N-cadherin in pretreated patients (*p* = 0.007). Regarding T2 signal, *RORC* was lower in T2-hypointense tumors of non-pretreated patients (*p* = 0.028); however, the number of cases was very low (n = 8). Moreover, there was a significant correlation between *RORC* and the percentage decrease in IGF-1 in patients presurgically treated with SRLs (Pearson’s r = 0.40, *p* = 0.007). In non-pretreated patients, the correlation was even stronger (Pearson’s r = 0.81) but was not statistically significant, since IGF-1 postsurgical reduction data for SRL monotherapy was only available for three cases. We did not find any association between the other genes and clinical variables.

### 3.4. GH-Producing Tumors Showed Different Intermediate EMT States with No Association with SRL Response

According to the SRL biochemical postsurgical response available for 43 patients, 16 patients (37%) were complete responders (CR), 10 (23%) were partial responders (PR), and 17 (39%) were considered nonresponders (NR). Unsupervised hierarchical clustering based on the expression of EMT genes in patients presurgically treated with SRLs separated tumors into several clusters with different expression patterns (Figure 6). Most of the tumors showed a hybrid or intermediate epithelial/mesenchymal phenotype, which indicated that as a group, somatotropinomas presented EMT features with different partial EMT states, confirming the biological heterogeneity of these tumors, and that EMT played a role in the heterogeneous nature of these tumors. However, the clustering was not associated with the different SRL response categories. No clustering was found related to tumor invasiveness either. Unsupervised hierarchical clustering was not performed with non-pretreated patients due to the low number of patients with an available postsurgical response (*n* = 5).

### 3.5. Association of SNAI1 and RORC Expression with SRL Response

From all markers, *SNAI1* expression was associated with SRL response categories in the whole cohort, while the association of *RORC* with SRL response was only found in the pretreated patients. Specifically, *SNAI1* expression presented an increasing trend from CR patients through PR to NR (*p* = 0.075) (Figure 7A), and NR patients had significant higher levels of *SNAI1* than CR (*p* = 0.025). When binomial logistic regression was constructed for phenotypes that benefited or did not benefit from SRL therapy (CR and PR vs. NR), *SNAI1* showed an AUC-ROC curve of 0.58 for a cut-off of 0.067, with a low sensitivity of 35.3% and a high specificity of 92.3% (*p* = 0.077) (Figure 7B).

The opposite pattern was found for *RORC* in the SRL-pretreated group (*p* = 0.003) (Figure 8A). Specifically, *RORC* expression was higher in CR compared to PR and NR (*p* = 0.051, and *p* < 0.001, respectively). The association of *RORC* expression with SRL response was not analyzed in the non-pretreated group due to the low number of cases with this information available (*n* = 5). However, we observed similarly low levels of *RORC* in the whole non-pretreated group and pretreated NR patients (*p* = 0.42), while pretreated CR and PR patients showed higher levels of *RORC* (*p* < 0.001 and *p* = 0.03, respectively). In addition, a categorical analysis of the normalized *RORC* expression in quartiles was performed to evaluate any nonlinearity in estimated effects. Furthermore, no further risk increase was found for *RORC* expression over the third quartile. We constructed a binomial logistic regression for phenotypes that normalized (CR) or did not normalize IGF-1 (PR and NR); the AUC-ROC curve that *RORC* showed was 0.81 for a cut-off of 1.2, with a sensitivity of 85.7% and a specificity of 76.9% (*p* = 0.016) (Figure 8B).

We also analyzed the protein expression of *RORC* in 27 tumors presurgically treated with SRLs (5 CR, 10 PR, and 12 NR) by immunohistochemistry. *RORC* was detected both in the cytoplasm and the nucleus. However, while all cases showed cytoplasmic staining, we observed that only 18% (5/27) of tumors presented a *RORC*-positive nuclear staining (Figure 9A) that was associated with SRL response, specifically in CR patients (χ^2^ test *p* = 0.031) (Figure 9B).

## 4. Discussion

EMT plays a fundamental role in the development of multiple tissues, including the pituitary gland [32,33]. This physiological process, which is aberrantly used by tumor cells, also occurs in pituitary tumors [34,35], especially in acromegaly [18,31], where EMT has been related to the response to SRLs [12,19,22]. Here, we further investigated EMT in GH-producing tumors, and found that most of them exhibited hybrid intermediate epithelial/mesenchymal expression profiles, which could be explained by the activation of alternative EMT programs and the progression of individual cells to different states along the EMT spectrum. This hybrid epithelial/mesenchymal, also called partial EMT, has been recently recognized to be the most common state in many solid malignancies, rather than a complete mesenchymal differentiation [36]. Furthermore, partial EMT is thought to enhance invasion and drug resistance in tumor cells [37].

The correlations we found in GH-producing tumors between EMT-related genes could denote a coordinated gene program for EMT in these adenomas as well, as it occurs in solid malignancies [38]. However, these correlations were modified by presurgical treatment with SRLs, which influenced the expression of some EMT-related genes, with *RORC* being the most affected one. Our findings in relation to *RORC* were in agreement with a previously published study that revealed an increase in *RORC* expression upon treatment with SRLs [31]. The fact that *RORC* behaved differently in patients naïve to treatment and in presurgically treated patients suggested that *RORC* expression may be regulated by different pathways. Of particular interest was that the overexpression of *RORC* observed in cases treated presurgically with SRLs correlated with a reduction in IGF-1 levels during postsurgical treatment, in agreement with a previous report [31]. Most importantly, we found that high *RORC* levels in GH-producing tumors may predict a complete response to SRLs, with an AUC of 81%, slightly better than E-cadherin [12], in SRL-pretreated patients. This was in line with our previous study showing that high levels of *RORC* could identify improved SRL response after surgical debulking [26]. Interestingly, our findings indicated the nonlinearity of *RORC* expression regarding SRL response, suggesting that SRL response is related to a specific expression level in which a therapeutic response is closer to a categorical behavior of *RORC*, rather than to a dose–response effect. Moreover, the nuclear immunostaining of *RORC*—specifically its functional orthotopic subcellular location [39]—was also associated with a better response to SRLs. Given that in daily clinical practice, a substantial proportion of patients are treated with SRLs before surgery, these findings could have important implications in improving the management of acromegalic patients. Although the response to presurgical SRL treatment could predict the expected response to postsurgical SRL therapy [40,41], this was not the case in our cohort [26], in which 17% of the patients changed their SRL response upon debulking; which highlighted the potential clinical value of *RORC*.

The *RORC* gene encodes for the RAR-related orphan receptor C, a ligand-regulated transcription factor, with known roles in immunological processes [42,43], circadian regulation [44], and hormone-signaling modulation in the thymus [45]. *RORC* has also been involved in the biology of certain cancer types, such as breast, renal, or pancreatic cancer [39,46,47,48]. Furthermore, *RORC* has been linked to TGF-β-induced EMT in hepatocytes during liver fibrosis, and in estrogen-receptor-negative breast cancer, and has been proposed as therapeutic target to treat these diseases [39,48,49]. In agreement with a previous study of healthy and tumoral skin, we found cytoplasmic and nuclear *RORC* immunostaining [50]. However, while all GH-producing tumors showed cytoplasmic expression, we found variation in the nuclear immunopositivity, as also occurs in melanoma. The fact that *RORC* is a nuclear receptor [51], and that its nuclear immunostaining in GH-producing tumors is associated with SRL response, established an interesting connection that suggested a functional role of *RORC* in the tumor nucleus. Therefore, further functional studies are warranted to elucidate the role of *RORC* in the response to SRLs. If *RORC* plays an active role in the mechanism of response to SRLs, this could open a new path in the treatment of acromegaly to improve SRL response, especially since an agonist of *RORC* has been used to treat refractory metastatic cancer [52].

We also found that *SNAI1* was associated with tumor invasion and poor SRL response. The relationship between *SNAl1* and invasiveness has been reported in other pituitary tumors [34]. However, as far as we know, the relationship between *SNAI1* and SRL response is reported here for the first time. Although our results suggested that *SNAI1* expression could only explain resistance to SRLs in a third of patients, its very high specificity (92.3%) providing a very useful clinical information of negative SRL response. *SNAI1* is a direct repressor of E-cadherin and a transcription factor with a key role in EMT modulation [53]; thus, it could be directly related to the E-cadherin loss reported in some GH-producing tumors. Certainly, this is not the only EMT-related mechanism involved in resistance to SRLs in acromegaly, but a negative EMT-related effect upon responsiveness to SRLs may have been present with variable intensity in some of the NR patients. Elucidation of the mechanistic of this phenomenon also would be of much interest, as targeted therapies aiming to deactivate it would re-sensitize GH-producing tumors toward SRL responsiveness [22].

DNA methylation, a repressive epigenetic mark [54], has been shown to be another mechanism of inhibition of E-cadherin transcription in many tumors, including pituitary tumors [55,56]. However, contrary to previous studies, our results showed no methylation of the E-cadherin promoter independently of gene expression levels. These contradictory results could be explained by the specific techniques used; we used bisulfite sequencing, a semiquantitative method that measures DNA methylation at single-CpG resolution [57], while previous studies were based on methylation-specific PCR (MSP), a qualitative technique [58]. Indeed, discrepancies between DNA-methylation methods have been reported in other studies; MSP in particular can overestimate DNA methylation [59], mainly due to its high sensitivity (one methylated allele in 1000 unmethylated alleles) [58] enabling to detect low levels of DNA methylation with little or no biological impact. Thus, the assessment of E-cadherin DNA methylation in large cohorts of acromegalic patients by using quantitative techniques should be addressed. In light of our results, this epigenetic mechanism does not appear to be active in GH-producing tumors, or should at least be questioned.

Altogether, our results added another layer to the heterogeneous biological nature of GH-producing tumors [60], significantly increasing the complexity of establishing useful predictive factors, and explaining that multiple markers, such as E-cadherin, *SSTR2* and Ki-67 [12], or *RORC* and *SNAI1*, instead of a single one, are necessary to consistently predict the response to SRLs. Thus, an improved and useful interpretation of the biologic state of GH-producing tumors in the clinical setting and, moreover, their response to SRLs, require further investigation, most likely aimed at setting and validating a multimarker analysis through biological systems methodology.

The present study had some weaknesses, such as a relatively limited number of cases, mostly regarding patients non-pretreated with SRLs. In addition, the inclusion of patients harboring macrotumors and tumors with extrasellar invasion might have led to some bias, although at the very end, these types of tumors generally require pharmacologic treatment after surgery, which made our results of clinical interest. There was also an enrichment of SRL-pretreated patients, representing a very extended practice in Spain in order to improve the outcome of the surgery. Undoubtably, SRLs can change the molecular profile of somatotropinomas, and, as our data suggested, SRLs altered the dynamics of the EMT process. However, thanks to these peculiarities, we were able to detect *RORC*’s association with SRL response. The strengths of our study were the detailed cohort characterization and the balanced representation of the different SRLs response categories.

In summary, our data further supported the occurrence of EMT in acromegaly and its relationship with SRL response. *RORC* and *SNAI1* expression offered dual and reciprocal information that may help medical treatment decision making in acromegaly, and await full validation with larger series of cases.

## Figures and Tables

**Figure 1 biomedicines-10-00460-f001:**
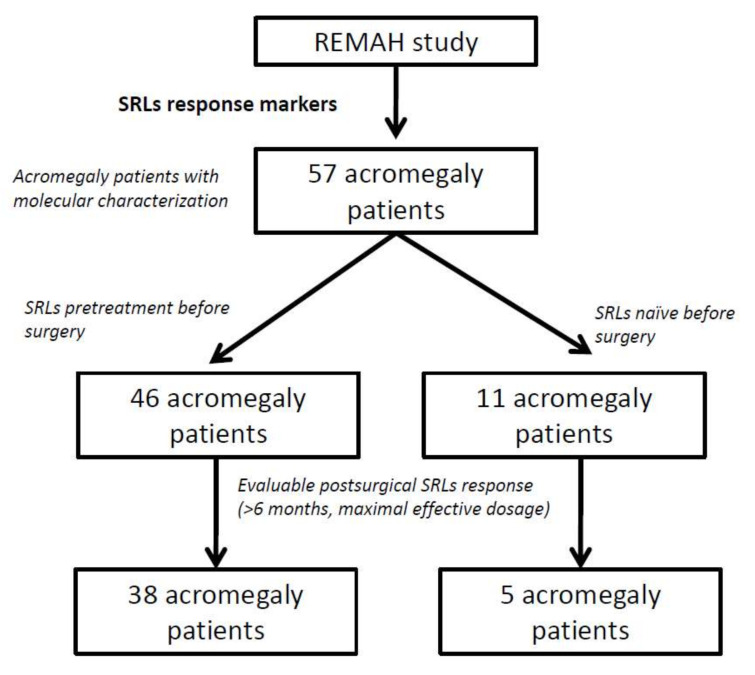
Scheme showing the cohort used in the study.

**Figure 2 biomedicines-10-00460-f002:**
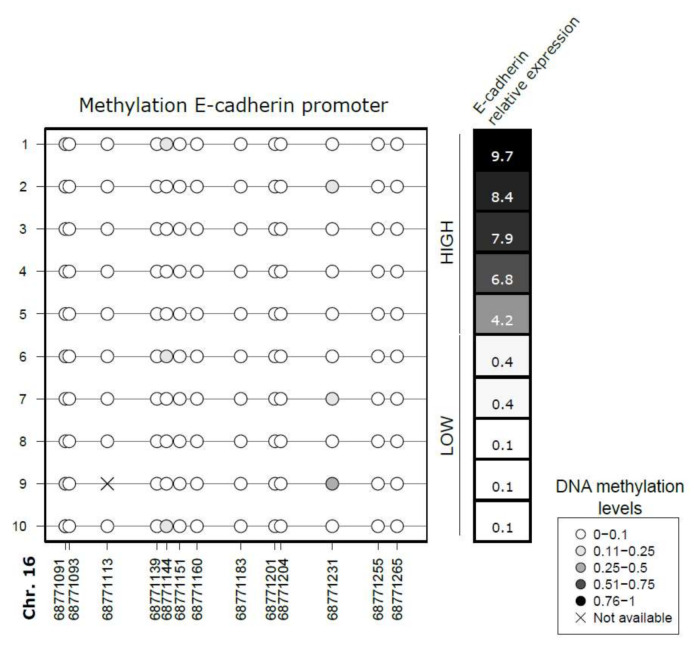
DNA methylation of E-cadherin promoter. Lollipop plot displaying the methylation levels of the CpG within the E-cadherin promoter sites as circles. The methylation levels were assessed by bisulfite sequencing in 10 somatotropinomas. We plotted the levels of DNA methylation (methylation ranges are indicated) and the levels of mRNA E-cadherin expression (indicated on the right) as a grayscale. The plot was performed using Methylation Plotter [28].

**Figure 3 biomedicines-10-00460-f003:**
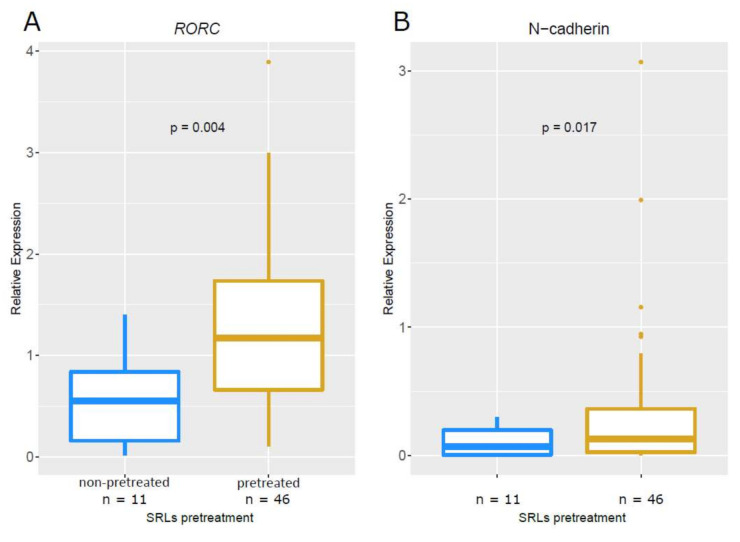
Effect of SRL presurgical treatment in *RORC* and N-cadherin expression. Boxplots showing relative expression of *RORC* (**A**) and N-cadherin (**B**) in patients treated presurgically with SRLs or not. In blue, patients not treated with SRLs before surgery; in dark gold, patients treated with SRLs before surgery.

**Figure 4 biomedicines-10-00460-f004:**
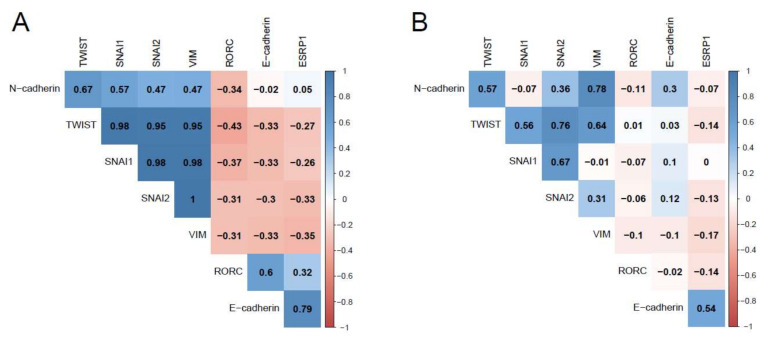
Correlations of EMT-related genes. (**A**) Pearson’s correlation matrix among the genes studied in the patients not treated with SRLs before surgery (*n* = 11). (**B**) Pearson’s correlation matrix among the genes studied in the patients treated with SRLs before surgery (*n* = 46). Pearson’s correlation coefficients are shown in the matrices; the intensity of color reflects the correlation magnitude.

**Figure 5 biomedicines-10-00460-f005:**
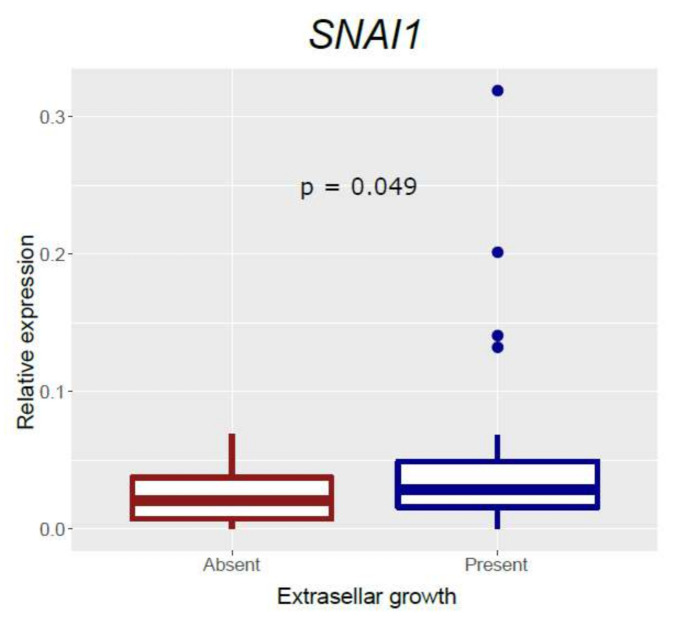
Boxplot showing relative expression of *SNAI1* according to tumor extrasellar growth. In dark red, tumors without extrasellar growth; in dark blue, tumors with extrasellar growth.

**Figure 6 biomedicines-10-00460-f006:**
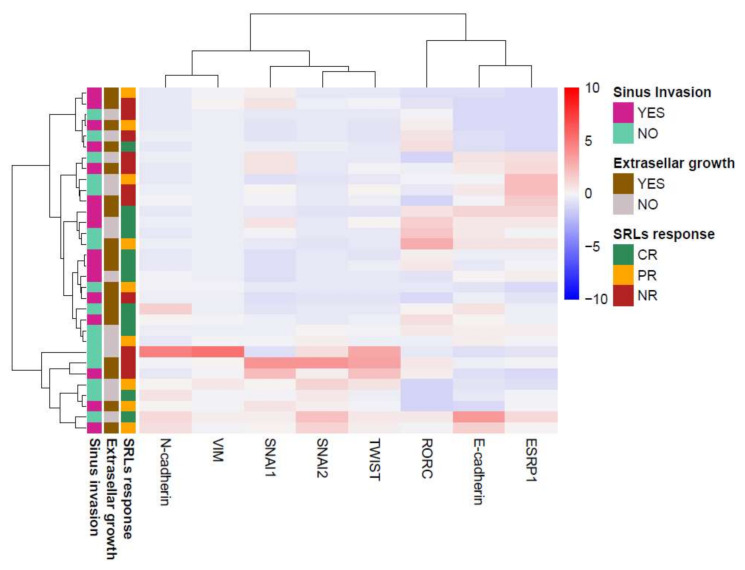
Dendrogram and unsupervised hierarchical clustering heatmap of the expression of the analyzed markers using Manhattan distance and Ward’s minimum variance method.

**Figure 7 biomedicines-10-00460-f007:**
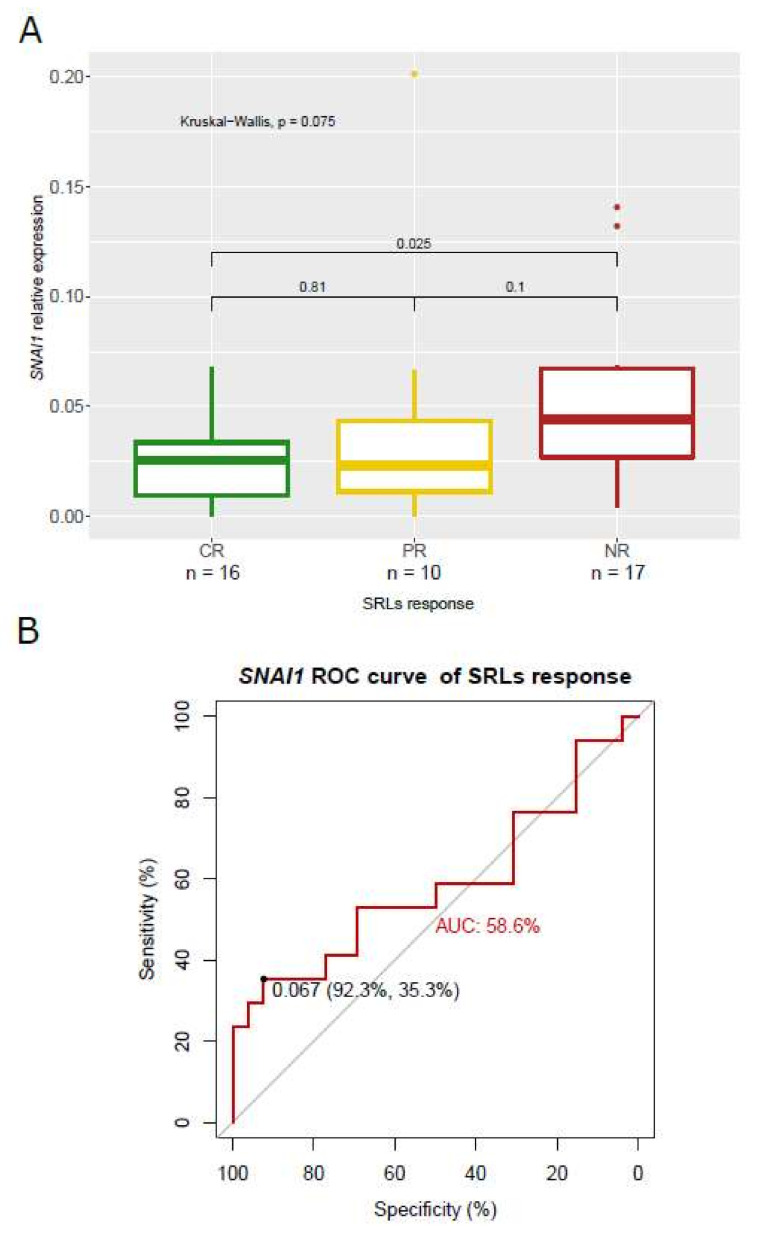
(**A**) Boxplot showing relative expression of *SNAI1* according to SRL response categories. Patients with CR, PR and NR were plotted in green, yellow and red, respectively. (**B**) ROC curve for *SNAI1* calculated with patients that benefited from SRL therapy (CR and PR) and patients that did not (NR).

**Figure 8 biomedicines-10-00460-f008:**
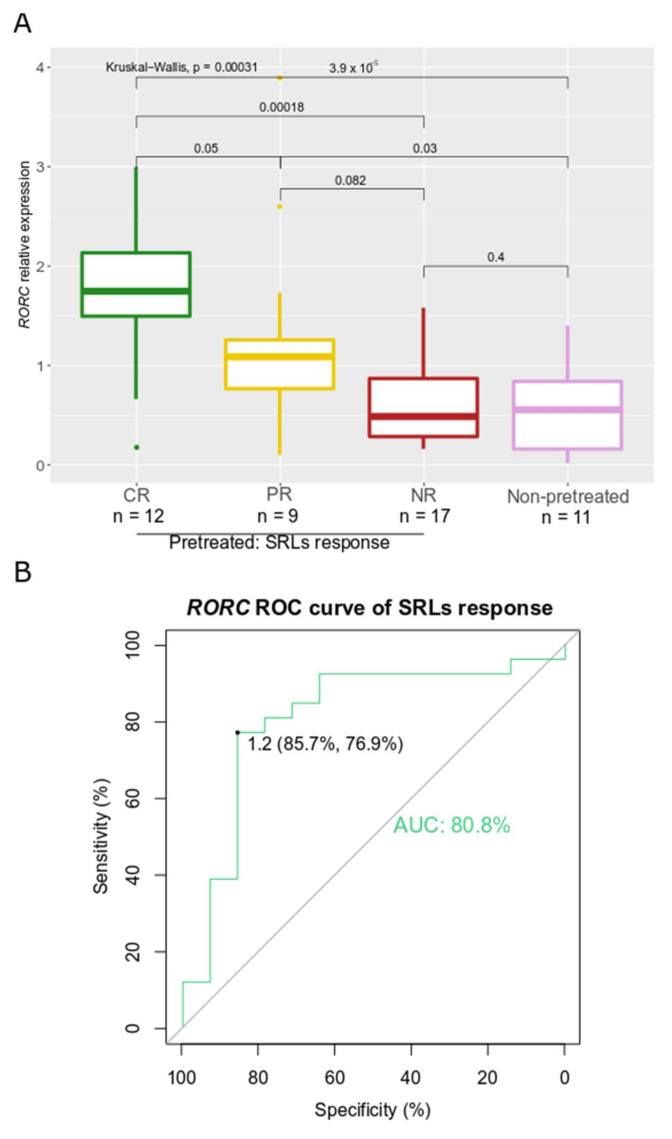
(**A**) Boxplot showing relative expression of *RORC* in patients treated with SRLs before surgery according to SRL response categories and in the whole group of non-pretreated patients. CR, PR, NR (all three pretreated patients) and non-pretreated patients plotted in green, yellow, red and purple, respectively. (**B**) ROC curve for *RORC* calculated with patients that normalized IGF-1 levels (CR) and patients that did not (PR and NR).

**Figure 9 biomedicines-10-00460-f009:**
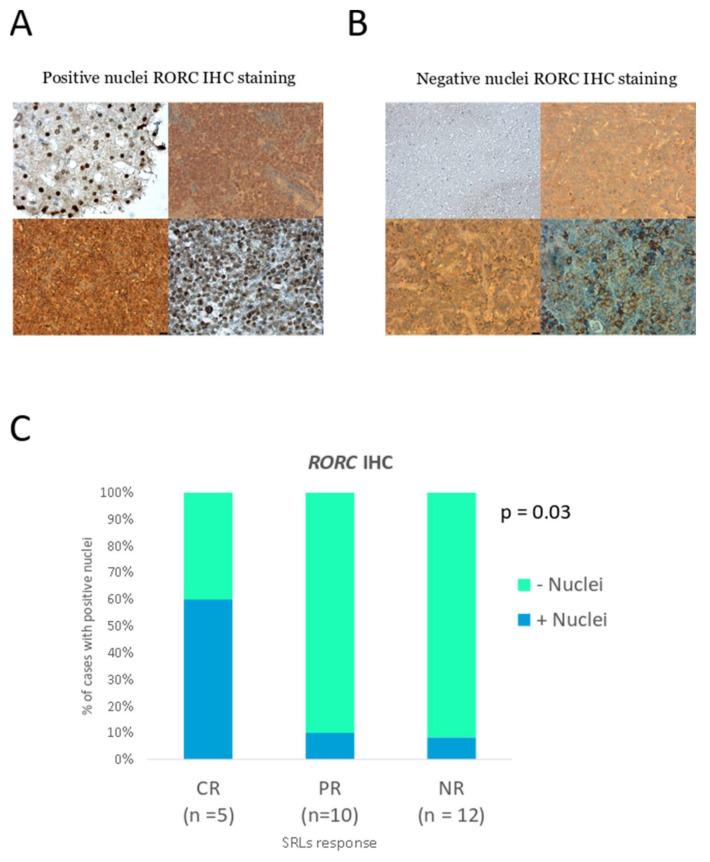
(**A**) Representative images of positive *RORC* nuclear immunostaining: From left to right and top to bottom, liver 40× (positive control), two GH-producing adenomas at 20× and one adenoma at 40×. (**B**) Representative images of negative *RORC* nuclear immunostaining. From left to right and top to bottom, brain control 20×, two GH-producing adenomas at 20× and one adenoma at 40×. The bar in the lower right corner indicates 25 µm (**C**) *RORC* immunohistochemistry (IHC) categorized according to nuclear positivity in complete responders (CR), partial responders (PR) and non-responders (NR) (*n* = 27).

**Table 1 biomedicines-10-00460-t001:** General and clinical characteristics of the patients and tumors included in the study.

Patients’ Characteristics		
Cohort (n)	57	
Male/Female	25/32	
Age, mean ± SD	45.74 ± 12.35	
Body mass index (BMI) ± SD	27.7 ± 5.21	
Medical Treatment		
SRL response	Pre * (46)	Non-pre ** (11)
Nonresponders	17	0
Partial responders	9	1
Complete responders	12	4
NA	8	6
Tumor Characteristics		
Macroadenoma	47 (82%)	
Extrasellar growth	39 (68%)	
Cavernous sinus invasion	27 (48%)	
Maximum tumor diameter (mm), mean ± SD	19.49 ± 10.03	
*GNAS* mutation	23 (41%)	
Hypointense T2 signal	19 (33%)	
Comorbidities		
Hypopituitarism	19 (33%)	
Visual alterations	12 (21%)	
Diabetes	17 (30%)	
Dislipemia	18 (31%)	
HBP	25 (44%)	
Cardiovasular incident	18 (32%)	

* Treated with SRLs before surgery; ** not treated with SRLs before surgery.

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
