# Peer review of "Implications of Heterogeneity of Epithelial-Mesenchymal States in Acromegaly Therapeutic Pharmacologic Response"

_biomedicines, 2022, doi:10.3390/biomedicines10020460_

Round 1

Reviewer 1 Report

The authors analysed the EMT gene expression in a cohort of 57 patients with acromegaly, including both the somatostatin analogue (SA) naïve patients and patients who received SA treatment before surgery. The tumours demonstrated a heterogeneous EMT gene expression pattern with the two genes, SNAI1 and RORC, appearing as predictive markers of the response to SA treatment. N-Cadherin and RORC gene expression were significantly affected by the preoperative SA treatment.

The manuscript is well written, and the results are of importance in the search for improved therapy of acromegaly patients. There are, however,  several suggestions for improvement of the manuscript.

  1. Patients' cohort. A majority of the patients had large and invasive tumours. Forty-six out of 57 patients received SA treatment before surgery.

The authors need to explain the criteria for selecting patients for preoperative SA treatment in paragraph 2.1 Patients.

If the patients were not randomised to direct surgery vs preoperative SA treatment, it should be discussed in more detail how potential selection bias affects the interpretation of the results.

In Table 1, extrasellar growth and sinus invasion need to be explained (cavernous sinus? paranasal sinus? which extrasellar structures were affected?)

Information on ethical approval should be given.

  1. DNA and RNA isolation (paragraph 2.2). The authors need to describe the type of specimens (fresh frozen or FFPE tissue) and explain how the presence of the representative tumour tissue in the specimens has been confirmed.
  2. RORC immunohistochemistry, Paragraph 2.5, Figure 9. If RORC should be used as a potential predictive marker, its expression needs to be assessable by using immunohistochemistry. There is strong intracytoplasmic staining in the microphotographs showing RORC expression, and nuclear expression is difficult to observe. According to the literature and the authors' description (p. 13, line 378), RORC is a nuclear protein. How do the authors interpret the intense intracytoplasmic staining? Is it background staining, or is there any evidence that the protein could be expressed in the cytoplasm? The authors should optimise the IHC protocol and avoid the cytoplasmic staining or increase the contrast between nuclear and cytoplasmic labelling so that positive nuclei can be easily observed? Description and microphotographs of the positive and negative control for RORC IHC should be provided as well as better microphotographs showing more distinct nuclear staining and no or less cytoplasmic background staining.
  3. In the last sentence of the abstract, the last part of the sentence, "avoid months of ineffective treatment", should be down-toned. The two markers might potentially improve the individualised treatment of patients with acromegaly, but it is still premature to state that they alone may help to "avoid months of ineffective treatment".

Minor comments:

  1. Page 7, line 229: Reference 30 is from 2013. Thus "recently" should be omitted.
  2. Page 8, line 269: "Few" should be removed.
  3. Page 12, line 344: What do the authors mean with "invasion drug resistance"? Should it be "invasion and drug resistance"?
  4. Page 13, line 368, the full stop punctuation mark after the brackets for ref. 26 should be removed.
  5. In the discussion section, p 13, lines 371 – 384, the abbreviation RORC is repeatedly followed by a symbol not previously used in the text. This needs to be explained if it is correct or removed if it is not correct.

Author Response

Response to Reviewer 1:

Dear reviewer,

We thank you for the careful examination of our work and for the constructive comments. Below, we respond to all issues raised, and indicate revisions that have been made to the manuscript.

With kind regards on behalf of the authors,

Prof. Manel Puig-Domingo

Dr. Mireia Jordà

The authors analysed the EMT gene expression in a cohort of 57 patients with acromegaly, including both the somatostatin analogue (SA) naïve patients and patients who received SA treatment before surgery. The tumours demonstrated a heterogeneous EMT gene expression pattern with the two genes, SNAI1 and RORC, appearing as predictive markers of the response to SA treatment. N-Cadherin and RORC gene expression were significantly affected by the preoperative SA treatment.

The manuscript is well written, and the results are of importance in the search for improved therapy of acromegaly patients. There are, however, several suggestions for improvement of the manuscript.

  1. Patients' cohort. A majority of the patients had large and invasive tumours. Forty-six out of 57 patients received SA treatment before surgery.

The authors need to explain the criteria for selecting patients for preoperative SA treatment in paragraph 2.1 Patients.

If the patients were not randomised to direct surgery vs preoperative SA treatment, it should be discussed in more detail how potential selection bias affects the interpretation of the results.

The reviewer is right that our cohort presents some bias. It was enriched in large and invasive tumors since we were studying SRLs response after surgery and thus this was an inevitable bias. Moreover, our cohort also suffers from enrichment in SRLs pretreated patients. This is because the pretreatment of patients a few months before the surgery is a very extended practice in Spain, in order to improve the outcome of patients. We agree that SRLs can undoubtfully change the molecular profile of somatotropinomas; however, we have been able to observe the relation of RORC and SRLs response due to this pretreatment.

We wrote an extended paragraph discussing these biases (page 14, lines 438-446):

The present study has some weaknesses, such as a relatively limited number of cases, mostly those regarding patients non-pretreated with SRLs. Also, the inclusion of patients harboring macrotumors and tumors with extrasellar invasion might lead to some bias, although at the very end, this kind of tumors are those generally requiring pharmacologic treatment after surgery which make our results of clinical interest. There is also an enrichment of SRLs pretreated patients which is a very extended practice in Spain in order to improve the outcome of the surgery. Undoubtably, SRLs can change the molecular profile of somatotropinomas, and, as our data suggest, SRLs alter the dynamics of the EMT process. However, thanks to these peculiarities, we were able to detect RORC’s association with SRLs response.

In Table 1, extrasellar growth and sinus invasion need to be explained (cavernous sinus? paranasal sinus? which extrasellar structures were affected?)

We thank the reviewer for the comment. In the revised version we have tried to be more descriptive regarding the extension of the tumours particularly, cavernous sinus invasion.

We added a more detailed explanation about extrasellar invasion (page 3, lines 105-110):

39 tumors (68%) presented extrasellar extension of the tumor: 3 patients (5%) presented only infrasellar extension, 20 (35%) presented only suprasellar extension and 16 (28%) presented both. Suprasellar invasion defined as a clear tumor growth through the diaphragma sella or above the plane of the inferior optic chiasm. Infra-sellar invasion was defined as clear tumor growth through the sellar floor and into the sphenoid sinus or clivus [27].

 Information on ethical approval should be given.

We wrote the following statement on this regard (page 15, lines 465-471):

The study was conducted in accordance with the ethical principles of the Declaration of Helsinki, the International Rare Diseases Research Consortium (IRDiRC) and implemented and reported in accordance with the International Conference on Harmonized Tripartite Guideline for Good Clinical Practice. The study was approved by the Germans Trias i Pujol Hospital Ethical Committee for Clinical Research (Ref. EO-11-080). The study protocol was approved by the research ethics board of each study site. This study was done under the auspices of the Spanish Molecular Registry of Pituitary Adenomas present in Orphanet [60].

  1. DNA and RNA isolation (paragraph 2.2). The authors need to describe the type of specimens (fresh frozen or FFPE tissue) and explain how the presence of the representative tumour tissue in the specimens has been confirmed.

We added an explanation regarding the types of samples of the study (page 4, lines 115-116).

Representative fragments of the fresh tumor were selected by a pathologist and embedded in RNAlater (Invitrogen) for 24h; after that, tumors were frozen at -80ºC. Histologic examination of that the frozen samples performed by a pathologist confirmed that they were part of the adenoma.

  1. RORC immunohistochemistry, Paragraph 2.5, Figure 9. If RORC should be used as a potential predictive marker, its expression needs to be assessable by using immunohistochemistry. There is strong intracytoplasmic staining in the microphotographs showing RORC expression, and nuclear expression is difficult to observe. According to the literature and the authors' description (p. 13, line 378), RORC is a nuclear protein. How do the authors interpret the intense intracytoplasmic staining? Is it background staining, or is there any evidence that the protein could be expressed in the cytoplasm? The authors should optimise the IHC protocol and avoid the cytoplasmic staining or increase the contrast between nuclear and cytoplasmic labelling so that positive nuclei can be easily observed? Description and microphotographs of the positive and negative control for RORC IHC should be provided as well as better microphotographs showing more distinct nuclear staining and no or less cytoplasmic background staining.

The reviewer raised a very interesting and important point regarding the RORC immunostaining. According to the Protein Atlas (https://www.proteinatlas.org/), RORC is expressed in the nucleus. However, some studies, like that from Slominski et al. 2014 published in the FASEB journal (PMID. 24668754), also showed a positive IHC of RORC in the cytoplasm. In this study, the authors performed IHC of normal skin and melanoma samples as well as cultured cells. Analysis of skin samples showed expression predominantly in the nucleus with weaker cytoplasmic expression. However, as occurs in our case, in melanoma, the staining of the cytoplasm remains while the nuclear immunopositivity varies. To check if this cytoplasmic staining was unspecific background or legitimate protein staining, they analyzed the protein fraction of the nucleus and the cytoplasm by western blot, and found the presence of RORC protein also in the cytoplasm.

Based on the images and results of the IHC in these articles, we believed that our IHC results are very similar to those from other authors and, therefore, it is difficult to believe that the staining is background specific of our technique.

We added the microphotographs that the reviewer proposed in the figure 9.

We commented on text (page 13, lines 390-396)

In agreement with a previous study in healthy and tumoral skin, we found cytoplasmic and nuclear RORC immunostaining [50]. However, while all GH-producing tumors showed cytoplasmic expression, we found variation in the nuclear immunopositivity as occurs in melanoma. The fact that RORC is a nuclear receptor [51] and its nuclear immunostaining in GH-producing tumors is associated with SRLs response establish an interesting connection suggesting a functional role of RORC in the tumor nucleus.

References:

Slominski AT, Kim TK, Takeda Y, Janjetovic Z, Brozyna AA, Skobowiat C, Wang J, Postlethwaite A, Li W, Tuckey RC, Jetten AM. RORα and ROR γ are expressed in human skin and serve as receptors for endogenously produced noncalcemic 20-hydroxy- and 20,23-dihydroxyvitamin D. FASEB J. 2014 Jul;28(7):2775-89. doi: 10.1096/fj.13-242040. Epub 2014 Mar 25. PMID: 24668754; PMCID: PMC4062828.

  1. In the last sentence of the abstract, the last part of the sentence, "avoid months of ineffective treatment", should be down-toned. The two markers might potentially improve the individualised treatment of patients with acromegaly, but it is still premature to state that they alone may help to "avoid months of ineffective treatment".

We changed the sentence and wrote:

SNAI1 and RORC may be useful to predict response to SRLs and help medical treatment deci-sion-making.

Minor comments:

We thank the reviewer deeply for detecting these errors.

  1. Page 7, line 229: Reference 30 is from 2013. Thus "recently" should be omitted.

The reviewer is right, we removed “recently”.

  1. Page 8, line 269: "Few" should be removed.

Done.

  1. Page 12, line 344: What do the authors mean with "invasion drug resistance"? Should it be "invasion and drug resistance"?

Exactly, we meant "invasion and drug resistance"

  1. Page 13, line 368, the full stop punctuation mark after the brackets for ref. 26 should be removed.

Done.

  1. In the discussion section, p 13, lines 371 – 384, the abbreviation RORC is repeatedly followed by a symbol not previously used in the text. This needs to be explained if it is correct or removed if it is not correct.
  2.  

Thank you for the observation. We removed the symbol. RORC have several aliases, but in the revised manuscript we have homogenized the nomenclature using the HUGO approved symbol and name which are RORC and RAR related orphan receptor C, respectively.

Reviewer 2 Report

The manuscript entitled "Implications of heterogeneity of epithelial-mesenchymal states in acromegaly therapeutic pharmacologic response" by Joan Gil and colleagues, aims to provide SRLs response predictors, related to EMT genes, for GH-producing pituitary tumors. Overall, it is an interesting study that provides further evidence about the relationship between EMT occurrence and SRLs response. Moreover, they confirmed the pivotal role of RORC and SNAI1 genes in the progression of GH-secreting tumors and their possible application for medical treatment of acromegaly. The work is well conducted, and the author’s suggestions are well supported by the experiments.

Minor suggestions:

1) Did the authors evaluated protein expression levels of E-cadherin (by immunostaining) in tumor tissues to confirm the mRNA analysis results?

2) In result 3 (SRLs treatment before surgery affects the expression of…) authors assessed that expression of vimentin, SNAI1, SNAI2 and TWIST1 doesn’t show any change after presurgical SRLs treatment. Negative data should be added to figure 3 o must be indicated “data no shown” in the text.

4) In figure 9, authors show one single representative image for both positive and negative RORC nuclear immunostaining; I suggest showing at least two representative images for both positive and negative staining with positive control tissue (as described in the text).

5) Please check the text for typos (e.g., line 93, octreotride instead of octreotide).

Author Response

Response to Reviewer 2:

Dear reviewer,

We thank you for the positive and constructive comments. Below, we respond to all issues raised, and indicate revisions that have been made to the manuscript.

With kind regards on behalf of the authors,

Prof. Manel Puig-Domingo

Dr. Mireia Jordà

The manuscript entitled "Implications of heterogeneity of epithelial-mesenchymal states in acromegaly therapeutic pharmacologic response" by Joan Gil and colleagues, aims to provide SRLs response predictors, related to EMT genes, for GH-producing pituitary tumors. Overall, it is an interesting study that provides further evidence about the relationship between EMT occurrence and SRLs response. Moreover, they confirmed the pivotal role of RORC and SNAI1 genes in the progression of GH-secreting tumors and their possible application for medical treatment of acromegaly. The work is well conducted, and the author’s suggestions are well supported by the experiments.

Minor suggestions:

1) Did the authors evaluated protein expression levels of E-cadherin (by immunostaining) in tumor tissues to confirm the mRNA analysis results?

We evaluated the E-cadherin protein levels by immunohistochemistry as described in our previous paper (Puig-Domingo et al., 2020). We did it for 34 patients included in our current manuscript, we found a significant Pearson’s correlation with mRNA expression (r=0.42, p=0.012).

2) In result 3 (SRLs treatment before surgery affects the expression of…) authors assessed that expression of vimentin, SNAI1, SNAI2 and TWIST1 doesn’t show any change after presurgical SRLs treatment. Negative data should be added to figure 3 o must be indicated “data no shown” in the text.

The reviewer is right. Sometimes we as researchers tend to focus on positive results only. We have explained in the revised manuscript that there were no differences between those genes (page 7, lines 240-242):

Expression of Vimentin, SNAI1, SNAI2, TWIST1 and ESRP1 did not showed significant differences between tumors presurgically treated with SRLs or naïve tumors (data not shown).

4) In figure 9, authors show one single representative image for both positive and negative RORC nuclear immunostaining; I suggest showing at least two representative images for both positive and negative staining with positive control tissue (as described in the text).

As the reviewer proposed, we added more representative images of both stainings as well as the positive and negative controls used in the IHCs.

5) Please check the text for typos (e.g., line 93, octreotride instead of octreotide).

Thank you for finding this typo, we have carefully reexamined the manuscript looking for typos.

Round 2

Reviewer 1 Report

I do not have additional comments.

Author Response

We thank you for the positive and constructive comments.

With kind regards on behalf of the authors,

Dr. Joan Gil